

# Benchmarking a targeted 16S ribosomal RNA gene enrichment approach to reconstruct ancient microbial communities

Raphael Eisenhofer[1], Sterling Wright[2] and Laura Weyrich[2,3,4]

[1] The Globe Institute, University of Copenhagen, Copenhagan, Denmark
[2] Department of Anthropology, Pennsylvania State University, University Park, Pennsylvania, United States
[3] Huck Institutes of the Life Sciences, Pennsylvania State University, University Park, Pennsylvania, United States
[4] School of Biological Sciences, University of Adelaide, Adelaide, Australia

Corresponding author
Sterling Wright, svw5689@psu.edu

## ABSTRACT

The taxonomic characterization of ancient microbiomes is a key step in the rapidly growing field of paleomicrobiology. While PCR amplification of the 16S ribosomal RNA (rRNA) gene is a widely used technique in modern microbiota studies, this method has systematic biases when applied to ancient microbial DNA. Shotgun metagenomic sequencing has proven to be the most effective method in reconstructing taxonomic profiles of ancient dental calculus samples. Nevertheless, shotgun sequencing approaches come with inherent limitations that could be addressed through hybridization enrichment capture. When employed together, shotgun sequencing and hybridization capture have the potential to enhance the characterization of ancient microbial communities. Here, we develop, test, and apply a hybridization enrichment capture technique to selectively target 16S rRNA gene fragments from the libraries of ancient dental calculus samples generated with shotgun techniques. We simulated data sets generated from hybridization enrichment capture, indicating that taxonomic identification of fragmented and damaged 16S rRNA gene sequences was feasible. Applying this enrichment approach to 15 previously published ancient calculus samples, we observed a 334-fold increase of ancient 16S rRNA gene fragments in the enriched samples when compared to unenriched libraries. Our results suggest that 16S hybridization capture is less prone to the effects of background contamination than 16S rRNA amplification, yielding a higher percentage of on-target recovery. While our enrichment technique detected low abundant and rare taxa within a given sample, these assignments may not achieve the same level of specificity as those achieved by unenriched methods.

## INTRODUCTION

Research into the human microbiome has intensified in the past decade because of its association with human health and disease (*Ley, 2010*; *Lloyd-Price et al., 2017*; *Turnbaugh et al., 2007*). Given the long-term association and co-evolution of these microbial

communities with humans (*Moeller et al., 2016*), understanding past human microbiomes and how they have changed through time may offer important medical insights (*Weyrich, 2021*). Applying high-throughput sequencing techniques to archaeological dental calculus have enabled researchers to explore the oral microbiomes of ancient human populations (*Adler et al., 2013*; *Warinner et al., 2014*; *Weyrich et al., 2017*; *Wright, Dobney & Weyrich, 2021*). Studies analyzing ancient DNA (aDNA) in dental calculus from various archaeological contexts have improved our understanding of the evolution of certain oral microbes (*Eisenhofer et al., 2020*; *Ottoni et al., 2021*; *Warinner et al., 2014*), the lifeways of our ancestors (*Fellows Yates et al., 2021*; *Weyrich et al., 2017*), and even past demographic events (*Eisenhofer et al., 2020*; *Ottoni et al., 2021*; *Quagliariello et al., 2022*). However, many challenges still remain in the field, such as characterizing the entire microbial composition of ancient oral microbiomes.

In modern microbiota studies, such as those part of the Human Microbiome Project (*Turnbaugh et al., 2007*), the amplification of the hypervariable 16S ribosomal RNA (rRNA) encoding gene is frequently used to generate microbial community profiles (*Caporaso et al., 2012*; *Sanschagrin & Yergeau, 2014*). This marker possesses both conserved regions, which are used to design broad-specificity primers, and hypervariable regions, which are useful for phylogenetic analysis (*Woese & Fox, 1977*). However, biases inherent to 16S rRNA amplicon sequencing are connected with DNA extraction, PCR amplification, sequencing, and subsequent sequence analysis (*Kennedy et al., 2014*; *Knight et al., 2018*). Furthermore, 16S rRNA amplicon sequencing is problematic for reconstructing ancient microbiomes because it preferentially amplifies fragments from contaminant sources rather than authentically ancient ones, which confounds microbial composition estimates (*Weyrich et al., 2017*; *Ziesemer et al., 2015*). In contrast, shotgun sequencing is mostly agnostic with respect to the length of DNA fragments and researchers can more easily examine the DNA damage profiles of a sample, making it the current 'gold' standard in paleomicrobiological research (*Orlando et al., 2021*; *Pochon et al., 2023*). As a result, several methodological and bioinformatic strategies have been developed to analyze ancient dental calculus microbiomes from shotgun data (*Pochon et al., 2023*; *Sarkissian et al., 2021*).

Despite the many benefits of shotgun sequencing, it has its own constraints. For instance, shotgun sequencing is more expensive than amplicon-based 16S rRNA approaches (*Ranjan et al., 2016*) and requires more computational resources to align sequences against reference databases containing genomic information (*Luo, Rodriguez-R & Konstantinidis, 2013*; *Eisenhofer & Weyrich, 2019*; *Velsko et al., 2018*; *Sekse et al., 2017*). A dedicated computer server with 1,500 GB of RAM was required for aligning shotgun metagenomic data against a database containing 47,713 prokaryotic reference genomes (*Eisenhofer & Weyrich, 2019*). While k-mer-based tools such as Kraken (*Wood & Salzberg, 2014*; *Wood, Lu & Langmead, 2019*) require less computational resources and can work with large databases (>100 GB), it is difficult to validate and authenticate the outputs from these tools. Furthermore, most taxonomic classification strategies from shotgun metagenomic data sets are dependent on the availability of reference genomes in databases, which limits findings to known taxa (*Klapper et al., 2023*; *Eisenhofer & Weyrich, 2019*;

*Pochon et al., 2023*). This finding is significant as *Eisenhofer & Weyrich (2019)* found that ~60% of DNA sequences of dental calculus samples, on average, generated from a shotgun sequencing alignment approach were unassigned when using alignment-based approaches on data generated with shotgun sequencing. Another limitation of shotgun sequencing may be more prominent for samples with poor aDNA preservation and have most of their DNA originate from environmental sources. This limitation has been documented in pathogen-focused studies which found that this methodology is prone to both false positives and false negatives (*Campana et al., 2014*). The presence of non-target DNA, such as human DNA, in shotgun data sets also poses another issue, as its inclusion in the taxonomic profiling step can influence microbial diversity estimates (*e.g.*, spurious mappings) (*Pochon et al., 2023*). Because shotgun sequencing also has difficulty in identifying low-abundant and rare taxa (*Lasa et al., 2019*), alternative strategies to reconstruct the microbial diversity in ancient dental calculus samples should be explored.

Applying hybridization capture (also referred to as sequence capture, target capture, or targeted sequence capture) to recover the 16S rRNA gene fragments from ancient dental calculus is one such new method to explore ancient microbial diversity. Hybridization capture refers to an enrichment methodology that utilizes a set of biotinylated DNA baits that are complementary to DNA sequences of interest. This method increases the proportion of specific, targeted DNA fragments within DNA libraries, followed by parallel sequencing (*Lasa et al., 2019*). *Noonan et al. (2006)* were the first to demonstrate that hybridization capture targeting the nuclear genome of a Neanderthal library could be efficient in recovering aDNA fragments. Since then, this method has continued to gain traction in aDNA research and has been highly effective in recovering full mitochondrial (*Ávila-Arcos et al., 2011*) and nuclear genomes (*Carpenter et al., 2013*) from ancient samples. This approach avoids some of the biases inherent to 16S rRNA amplicon sequencing, such as being able to target short DNA fragments (<100 bp). Moreover, it typically recovers a higher proportion of endogenous DNA content from an aDNA library than shotgun sequencing (*Carpenter et al., 2013*; *Mohandesan et al., 2017*). In some microbiome studies, hybridization capture can also provide greater phylogenetic resolution and increase sensitivity at low sequencing depths than shotgun sequencing (*Lasa et al., 2019*; *Barrett et al., 2020*). Potentially, this approach can aid in reconstructing the microbial diversity of dental calculus samples because it would leverage the greater diversity present in 16S rRNA gene databases compared to genome databases. For example, the SILVA SSU (small subunit) Ref NR (non-redundant) 132 database contains 695,171 16S rRNA gene sequences (even after clustering at 99% sequence identity), compared to the 47,713 reference genomes tested in *Eisenhofer & Weyrich (2019)*. Furthermore, data from this approach would likely not require nearly as much RAM as data generated from shotgun sequencing (*e.g.*, >3 TB) (*Ranjan et al., 2016*). In *Eisenhofer & Weyrich (2019)*, more than a terabyte of RAM was needed to perform the metagenomic profiling step with the alignment-based MALTn program. While hybridization capture bait sets have been designed for a variety of microbial projects, including for modern (*Beaudry et al., 2021*) and ancient samples (*Bos et al., 2011*; *Schuenemann et al., 2011*; *Ziesemer et al., 2019*), a comprehensive evaluation of the efficacy

of employing hybridization capture to target the 16S rRNA fragments of oral microbes from archaeological dental calculus samples has yet been formally investigated.

Here, we develop and benchmark an *in silico* and experimental framework for performing a 16S rRNA enrichment analysis on ancient dental calculus samples. First, we investigate whether the aDNA characteristics impact taxonomic assignments. Second, we investigate whether different hybridization temperatures impact the efficacy of our hybridization enrichment approach. Lastly, we compare the efficacy of using a 16S rRNA hybridization enrichment capture approach to 16S rRNA amplicon and shotgun sequencing approaches (Fig. 1). Our results suggest that shotgun sequencing is still the most optimal approach for surveying the taxonomic diversity of samples and provides the highest resolution of data. Nevertheless, our results suggest that hybridization capture can recover rare or low abundant taxa that are missed in shotgun sequencing approaches. Therefore, future research may want to consider incorporating hybridization capture into their study design to improve their surveillance of ancient oral microbiomes.

## MATERIALS AND METHODS

### 16S rRNA gene RNA probe design

Full-length 16S rRNA genes were obtained from all species identified from a previous ancient dental calculus study (*Weyrich et al., 2017*) from the Ribosome Database Project (RDP) (*Cole et al., 2014*). To further increase diversity, 16S rRNA genes from species in the Human Oral Microbiome Database (HOMD) (*Chen et al., 2010*) that were not identified by *Weyrich et al. (2017)* were also included. This added 285 sequences, which resulted in 570 full-length 16S rRNA genes to be used for probe design (FigShare DOI: 10.25909/5cc11894b0cc2). RepeatMasker (*Smit, 2004*) removed simple and low-complexity repeats from the sequences. A total of 80 bp (base pair) RNA baits with 20 bp probe spacing and 4× tiling density were synthesized by Arbor Biosciences (formerly MyBaits). Baits were collapsed if they had fewer than 11 mismatches between each other, yielding a total of 19,634 80 bp RNA baits. While the probe design was based on 16S rRNA genes from ancient and modern oral taxa, the redundancy built into the probes should also capture microbial diversity that was not used as input for the probe sequences.

### Hybridization enrichment and DNA sequencing

A total of four samples that were previously published in *Weyrich et al. (2017)* using unenriched, whole-genome alignment approach (UeWGA) methods were selected for hybridization enrichment (Table S1). In brief, each library was created by amplifying existing library DNA in four 25 μL PCR reactions (each containing: 13.625 μL dH$_2$O, 2.5 μL 10X AmpliTaq Gold Buffer, 2.5 μL of 25 mM MgCl$_2$, 0.625 μL of 10 mM dNTPs, 2.5 μL of 10 μM forward and reverse primer, 0.25 μL AmpliTaq Gold, 3 μL template DNA) with the following PCR conditions: denaturing at 94 °C for 12 min before 13 cycles of (30 s denaturing at 94 °C, 30 s annealing at 60 °C, 45 s extension at 72 °C) followed by a final extension of 10 min at 72 °C (*Eisenhofer, 2018*). PCR amplifications were pooled and then cleaned with AMPure XP beads to reach 100 ng of DNA for input into hybridization enrichment. A modified version three of the MyBaits protocol was used for hybridization

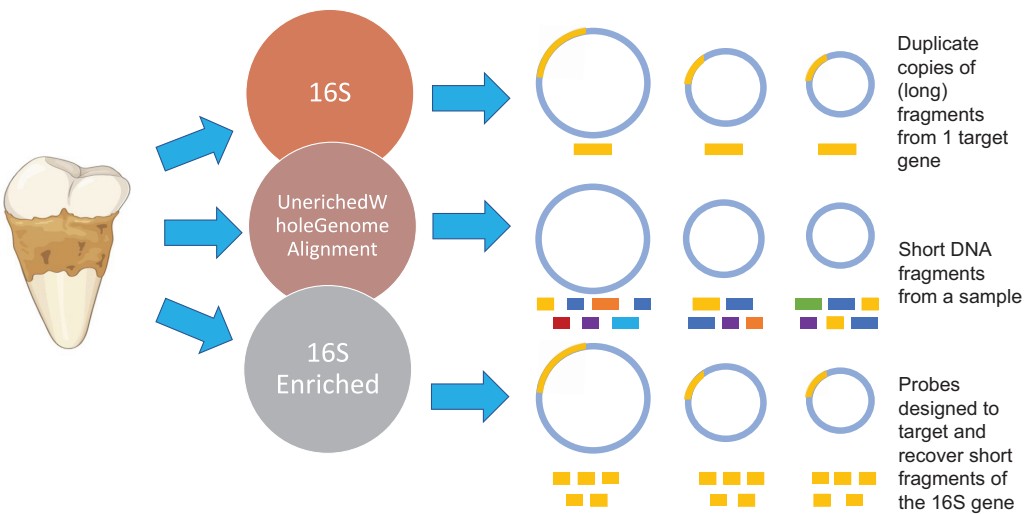

**Figure 1 Methodology overview.** Overview of the microbial characterization strategies. For the 16S samples, OTUs were clustered using UCLUS at 97% and taxonomically identified using Greengenes 12.10 database. Samples prepared using the UeWGA method, were prepared *via* shotgun sequencing and aligned with MALTn. For the 16S enriched samples, MyBaits probes were designed based on the oral taxa in the *Weyrich et al. (2017)* study and the Human Oral Microbiome Database. Each circle represents a genome of a microbe. The gold bars represent 16S genes of microbes. The different color bars for the Unenriched Whole Genome Alignment (UeWGA) method represent DNA fragments from different parts of the whole genome. For the 16S Enriched method, the gold bars also represent 16S fragments but are consistently shorter than the 16S rRNA gene amplicon approach.

enrichment, whereby the input RNA bait concentration was reduced to 25% of the recommended amount, and custom oligonucleotides were utilized to block the P5/P7 library adapters. To test the efficiency of hybridization at different temperatures, four samples were enriched at 55 °C and 65 °C for 40 h. Based on these results, the rest of the samples were enriched at 65 °C for 40 h. Following hybridization enrichment capture, the streptavidin beads were washed three times using Wash Buffer 2 (MyBaits v3 manual) and resuspended in a PCR mastermix (as above) before amplification with 13 cycles of PCR. Amplified libraries were cleaned with AMPure, quantified with an Agilent TapeStation, pooled at equimolar concentrations, and sequenced on an Illumina HiSeq X Ten platform (2 × 150 bp). Sequencing data for the samples processed with the hybridization capture method can be found on the Sequence Read Archive (https://www.ncbi.nlm.nih.gov/sra) under the accession number: PRJNA1031168.

## Sequence data processing and 16S rRNA gene enrichment analysis

The sequencing data was converted into the fastq format using Illumina's bcl2fastq software and were demultiplexed using AdapterRemoval2 based on unique P5/P7 barcode combinations (*Schubert, Lindgreen & Orlando, 2016*). We reduced the computational time of downstream analyses, seqtk (https://github.com/lh3/seqtk) was used to randomly subsample each 16S rRNA enriched library to 100,000 reads. SortMeRNA (*Kopylova, Noé & Touzet, 2012*) was used with the SILVA bacterial and archaeal 16S rRNA databases (SILVA 119, 95% clustered) to identify 16S rRNA gene fragments from samples.

We constructed a BLAST database (*Altschul et al., 1990*) using the SILVA 128 NR99 16S rRNA reference database (*Quast et al., 2012*) to assign taxonomic identifications for the putative 16S rRNA gene fragments. The putative 16S rRNA gene fragments were aligned to the database using the default parameters, with the following exceptions: –evalue = 0.01 for added stringency and –outfmt = 0 to allow for import into MEGAN. We tested the impact of missing reference sequences on taxonomic classification by creating a filtered SILVA database and removed reference sequences of taxa from of our simulated dataset using the Filterbyname.sh script from BBtools (https://jgi.doe.gov/data-and-tools/bbtools/). We used grep to create a list of taxa to remove from the reference sequences in the SILVA database using Filterbyname.sh (Table S2). Lastly, BLAST outputs were then imported into MEGAN CE (v. 6.24.15) (*Huson et al., 2016*) using the "Import from BLAST" option with the synonyms mapping file (SSURef_NR99_128_tax_silva_to_NCBI_synonyms.map.gz) obtained from the MEGAN community website: (https://software-ab.cs.uni-tuebingen.de/download/megan6/welcome.html).

### Generating simulated dataset with ancient DNA characteristics

To test if our 16S rRNA gene fragment analysis pipeline could accurately reconstruct microbial communities, we constructed a simulated dataset using Gargammel (*Renaud et al., 2017*). 16S rRNA genes from 19 phylogenetically diverse prokaryotic species were obtained from the SILVA 128 NR99 16S rRNA database (*Quast et al., 2012*) (Table S2) and randomly fragmented to create 100,000 16S rRNA gene fragments fitting a log-normal aDNA fragment-length distribution (Fig. S1) (gargammel -n 100,000 –loc 4 –scale 0.3). Varying levels of cytosine deamination (aDNA damage) were simulated using deamSim from gargammel on our simulated dataset to create three different levels of simulated single-stranded overhang deamination: 10% (low) deamination (-damage 0.03, 0.25, 0.01, 0.1); empirical (moderate) deamination from a published mapDamage profile (*Jónsson et al., 2013*; *Olalde et al., 2014*); and 50% (high) deamination (-damage 0.03, 0.25, 0.01, 0.5). Lastly, the resulting simulated 16S rRNA gene datasets were mapped against the SILVA 128 NR99 database and the filtered SILVA database using BLAST, as described above.

### Taxonomic classification of dental calculus data processed with the UeWGA pipeline

Sequencing data bioinformatically processed with the UeWGA pipeline come dental calculus samples that were previously published (NCBI SRA database under BioProject PRJNA685265) and steps for this pipeline are described in *Eisenhofer & Weyrich (2019)*. In brief, reads were aligned against a NCBI microbial reference database containing 47,713 genomes from bacteria and archaea using MALTn (*Herbig et al., 2016*) with default parameters and outputting BLAST text files. The LCA parameters used for the UeWGA data were: bitscore = 50, E-value = 0.01, minsupp = 0.1 (*i.e.*, a taxonomic assignment requires at least 0.1% of reads to pass), and the weighted LCA algorithm (80%), as suggested in *Huson et al. (2016)*. The resulting BLAST text files were converted into RMA6 files using the blast2rma script in MEGAN. Operational Taxonomic Units (OTUs) resulting from a 16S rRNA amplicon approach on the same samples were also obtained

from a previous study (*Weyrich et al., 2017*). For the 16S samples, OTUs were clustered using UCLUST at 97% and taxonomically identified using the Greengenes 12.10 database.

## Comparison of 16S rRNA gene enrichment, 16S rRNA amplicon datasets, and UeWGA datasets

LCA parameters used for the 16S enrichment method were the same as the ones in the UeWGA dataset except, except that the naïve LCA algorithm (80%) was used because false-positive taxonomic assignments were identified in the simulated 16S rRNA gene fragments with the weighted algorithm. 16S rRNA amplicon data was directly imported into MEGAN (v. 6.24.15) (*Huson et al., 2016*).

## Statistical analyses

Feature-level (*i.e.*, sequences assigned to the lowest taxonomic rank) and genus-level assignments in MEGAN were exported as BIOM tables and then imported into QIIME2 (v.2021.11) (*Caporaso et al., 2012*). Samples with less than 1,000 counts were excluded from downstream analyses. For the alpha and beta diversity analyses, unfiltered data sets were compared to ascertain the differential impacts of contaminant taxa. For comparisons of specific taxa across methods, genera and species identified in the extraction blank controls were removed from biological samples in MEGAN. When calculating beta diversity using Bray Curtis, the counts of all samples were rarefied to the sampling depth for the sample with the lowest number of feature-level (9,980 for A12017 EuroHG2) and genus-level assignments (9,731 for A13208_AfrSF2). Alpha diversity measures were calculated based on the number of 'observed features' and 'observed genera' using the alpha-group-significance plug-in (Kruskal-Wallis test). Phyloseq was also used to perform a rarefaction curve analysis using different alpha diversity indexes (Shannon, Chao1, and total observe features/genera). Bray Curtis distance matrices at the feature and genus level were computed with the qiime diversity core-metrics plugin and used for the Principal Coordinate Analysis (PCoA) and PERMANOVA tests in QIIME2. Aitchison distances were computed with the qiime deicode rpca plugin and used for the PCoAs visualizations and PERMANOVA tests in QIIME2.

## RESULTS

### Assessing taxonomic assignment of simulated ancient 16S rRNA gene fragments

We created a simulated dataset containing 19 phylogenetically diverse prokaryotic species to test how accurately we could classify short 16S rRNA gene fragments (Table S2). The 16S rRNA gene from each species was selected and randomly fragmented to fit a commonly observed aDNA fragment length distribution (*i.e.*, log normal with a mode of 50 bp; Fig. S1). DNA fragments were classified taxonomically by mapping DNA fragments to the SILVA 16S rRNA database (NR 99, release 128) using BLAST nucleotide alignment, following by applying the default lowest common ancestor (LCA) algorithm in MEGAN6 (v. 6.24.15). We found that 98.4% of the total DNA fragments were assigned taxonomically (Fig. 2A), and of these, 53.2% were assigned to the genus level (Fig. S2), and 3.5% to the

species level (Table 1; Fig. S2). The resulting genus-level taxonomic composition matched that of the input sequences with one exception: *Yersinia* 16S rRNA gene fragments could not be assigned to the genus *Yersinia* and were pushed up to the order Enterobacterales (Fig. 2B). The 1.6% of fragments that were not aligned were extremely short (between 15–30 bp) (Fig. S1), suggesting that accurate classifications are limited to at least 30 bp for 16S rRNA fragments, as previously observed for whole-genome alignments (*Eisenhofer & Weyrich, 2019*).

To test if cytosine deamination (a characteristic of aDNA) influences taxonomic classification of short 16S rRNA gene fragments, we simulated three levels of sequence deamination on our simulated dataset representing low (10%), moderate (simulated from a real mapDamage profile from an ancient specimen; –30% (*Olalde et al., 2014*); and high (50%). We simulated moderate damage with the La Brana individual because the damage profile for this sample has been benchmarked in previous simulation studies and represents a typical ancient sample (*Oliva et al., 2021*). We found that deamination did not have a notable impact on taxonomic classification, as there was no loss of taxonomic assignments or misclassifications (Figs. 2E–2G). Only a 1.5% loss of reads assigned taxonomically was observed when assessing the highest level of deamination (50%) compared to the non-deaminated simulated dataset (Table 1). Overall, these findings suggest that deamination does not meaningfully hinder the ability to classify ancient 16S DNA fragments and that the method developed here could be used to examine taxonomic diversity even in highly, degraded ancient samples. Our results are consistent with prior research, indicating that deamination has only a marginal effect on taxonomic classification for data generated with shotgun sequencing (*Eisenhofer & Weyrich, 2019*; *Velsko et al., 2018*; *Warinner et al., 2017*).

To test the impact of missing reference sequences on our ability to taxonomically classify 16S rRNA gene fragments, we performed a species and genus exclusion experiment whereby we removed reference sequences corresponding to species or genera present within the simulated dataset from the SILVA database (Table S2), and then repeated BLAST alignments using this modified database. Sequences originating from excluded species could be classified to their respective genera (Fig. 2C, Table 1). Similarly, removal of all reference sequences attributed to the *Streptococcus* genus (which accounts for 25% of the simulated dataset) resulted in a 12% reduction in assignments to the genus-level (53.2% *vs*. 41.2%), and an associated increase in sequences at higher taxonomic ranks (Fig. 2D, Table 1). Importantly, the total number of aligned reads did not substantially decrease (97.7% *vs*. 98.4% for non-exclusion) (Fig. S2), suggesting that sequence conservation and phylogenetic signal within the 16S gene allowed for placement of these sequences higher in the taxonomy, rather than discarding them outright.

## Optimizing hybridization enrichment efficiency

We designed RNA baits to capture 16S rRNA gene fragments from a diverse range of microbial taxa. We tested whether the proportion of on-target sequences and non-target sequences impacted by differing hybridization temperatures, specifically 55 °C and 65 °C. We found that enrichment at 65 °C yielded the highest on-target enrichment of 16S rRNA

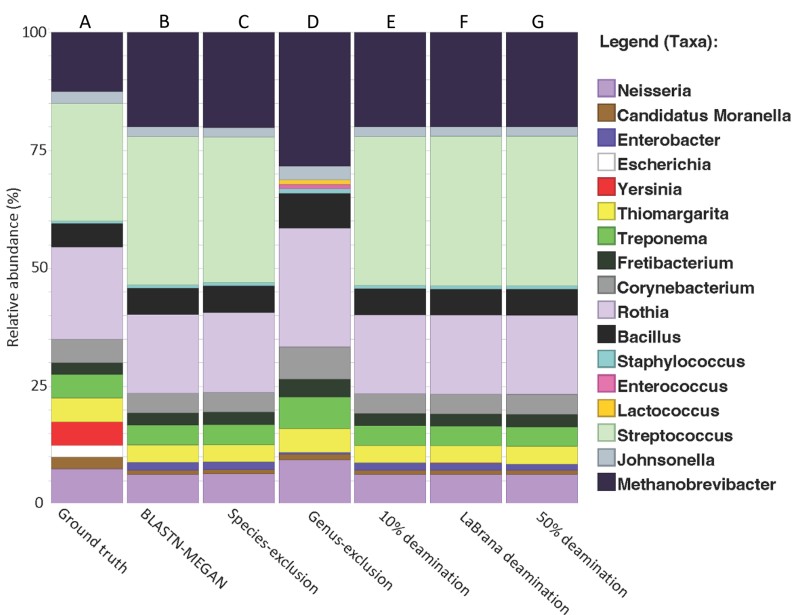

**Figure 2** **Recapitulation of genus-level taxonomic composition from simulated metagenome.** (A) Actual abundance and structure of simulated Community. (B) Reconstruction of community structure using MALT/MEGAN. (C and D) Results of genus and species exclusion test. (E–G) Results of different rates of simulated cytosine deamination on metagenomic reconstruction.

gene fragments (average 53.42% putative 16S rRNA gene fragments). This was more than a two-fold increase over the 55 °C enrichment (average 26.17% putative 16S rRNA gene fragments), and a 334-fold increase over the unenriched shotgun libraries (average 0.16% putative 16S rRNA gene fragments) (Table S3). We found that increasing hybridization temperature selected for longer mean DNA fragment lengths (unenriched = 56 bp, 55 °C = 69 bp, 65 °C = 79 bp), but did not find changes in the mean GC content between temperatures (Table S4). This enrichment of longer DNA fragments by higher temperatures could be explained by shorter DNA fragments not having sufficient Watson-Crick hydrogen bonding sites to remain attached to probes at higher temperatures, resulting in the preferential binding and enrichment of longer DNA fragments.

## Hybridization temperature minimally influences diversity

Given that hybridization temperature was found to select for longer DNA sequences, we next tested to see if this influences the alpha and beta diversity of samples. Due to constraints in the alignment speed of BLAST, each sample from each treatment (55 °C, 65 °C) was randomly subsampled to 100,000 reads. After alignment and classification using the SILVA database, genus and feature level assignments were exported separately from MEGAN6 (v.6.24.15) into QIIME2 (v.2021.11). The alpha diversity between the 55 °C and 65 °C groups were insignificant at the feature level (Kruskal-Wallis; $p = 0.77$) and genus level (Kruskal-Wallis; $p = 1$) (Table S5). Beta diversity between the two groups were also similar (PERMANOVA; $p > 0.05$) when using either Bray-Curtis and Aitchison

**Table 1 Alignment statistics for simulated data.**

| Percentage of total reads assigned | Total | Domain | Phylum | Class | Order | Family | Genus | Species |
|---|---|---|---|---|---|---|---|---|
| BLASTN-MEGAN | 98.4% | 4.5% | 3.7% | 11.1% | 4.8% | 17.0% | 53.2% | 3.5% |
| BLASTN-MEGAN-SpeciesExclusion | 98.3% | 4.6% | 3.9% | 11.1% | 4.9% | 17.1% | 55.4% | 0.7% |
| BLASTN-MEGAN-GenusExclusion | 97.7% | 9.6% | 6.7% | 12.0% | 7.4% | 18.8% | 41.2% | 0.2% |
| BLASTN-10%Deamination | 97.8% | 4.9% | 3.8% | 11.2% | 4.8% | 16.9% | 52.7% | 3.4% |
| BLASTN-MEGAN-LaBrana-Deamination | 97.8% | 5.1% | 3.8% | 11.1% | 4.8% | 16.9% | 52.6% | 3.4% |
| BLASTN-MEGAN-50%Deamination | 96.9% | 5.5% | 3.7% | 11.2% | 4.9% | 16.9% | 52.1% | 3.4% |

distances at the genus and feature levels (Figs. 3A and 3B; Table 2). Regarding taxonomic assignments specific to hybridization temperature, taxonomic bar plots representing the sum of samples per treatment were nearly identical in both abundance and diversity (Fig. S3). Both enrichment temperatures allowed for the detection of *Corynebacterium* (a genus commonly found in modern human dental plaque) and was not identified in unenriched samples. The 65 °C treatment was also the only one to detect *Pseudoramibacter* (an oral taxon previously identified in ancient dental calculus and is present in modern oral studies (*Eisenhofer & Weyrich, 2019*; *Siqueira & Rôças, 2003*)) (Fig. S4). Overall, these results suggest that a hybridization enrichment can be used to identify microbial taxa within dental calculus and that temperature of 65 °C may provide a mild improvement on the recovery of 16S rRNA gene fragments. Therefore, a hybridization temperature of 65 °C was used for subsequent enrichments.

## Differences in diversity are observed between 16S rRNA enrichment, whole-genome alignment, and 16S rRNA amplification methods

We next sought to examine if the diversity and composition of communities obtained from 16S rRNA gene enrichment were distinct from the UeWGA and 16S rRNA amplified strategies. We compared 16S rRNA enrichments on samples with previously published UeWGA and 16S rRNA amplification data (*Eisenhofer & Weyrich, 2019*; *Weyrich et al., 2017*). We enriched 11 additional dental calculus samples from *Weyrich et al. (2017)* at 65 °C, bringing the total number of samples to 15 (including four samples from the previous section).

Alpha-rarefaction curves indicated that sampling completeness for both feature-level and genus-level was reached at a sequencing depth of 1,000 reads (Shannon's Diversity) and >3,000 sequences (observed taxa) for 16S enriched, 16S rRNA amplicon, and UeWGA, (Fig. S5). Group significance for alpha diversity was calculated using the Kruskal-Wallis test for both observed features and genera BIOM tables. Results indicate that UeWGA recovered more taxa at the feature and genus level than both the 16S hybridization enrichment and amplicon strategies (Figs. 4A and 4B). The alpha diversity among the three treatments were significantly different when using feature ($p = 0.00000009$; $H = 32.372$) and genus-level assignments ($p = 0.002$; $H = 16.923$) for each of the metrics examined ($q < 0.05$; Table S6). Each pairwise comparisons among the three groups were also significant at the feature and genus level ($q < 0.05$; Table S6) across all three groups,
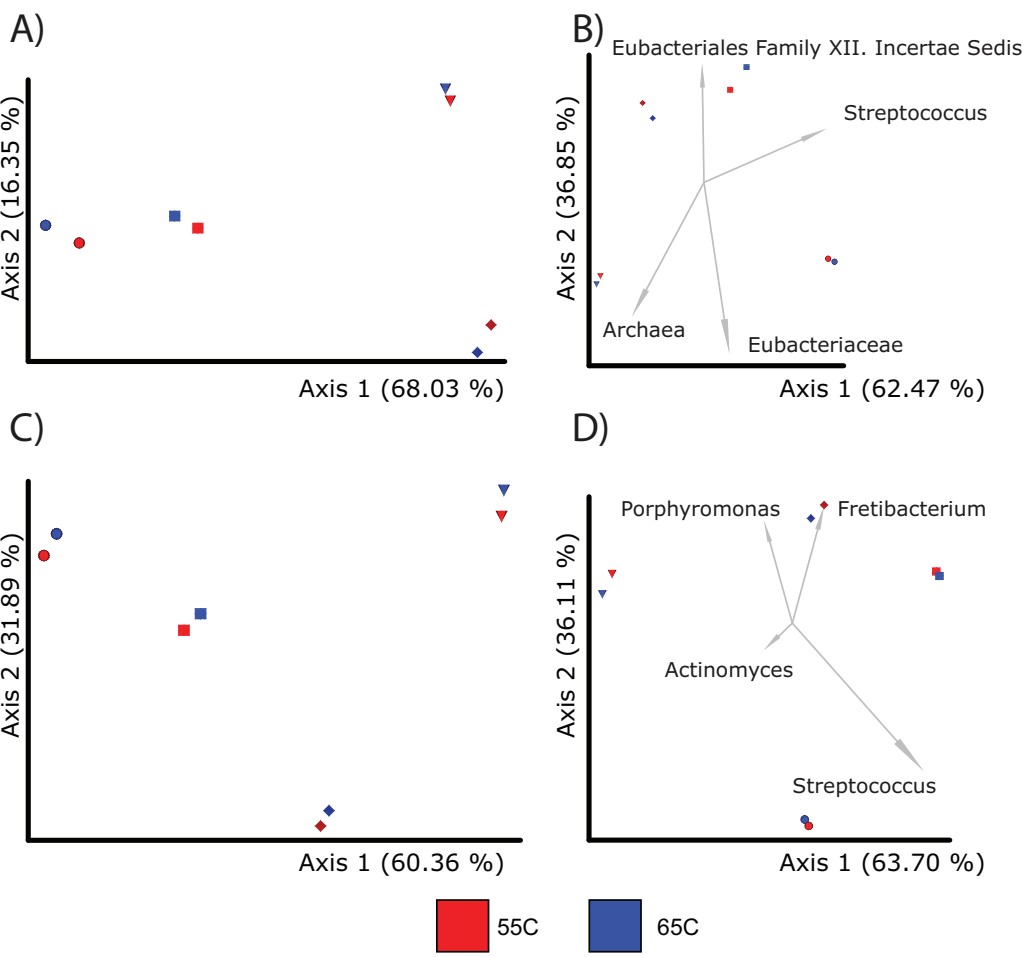

**Figure 3** **PcoA plots comparing the composition of microbial communities recovered from samples using different enrichment temperatures.** Each sample is color coded to their respective temperature (55 °C = red and 65 °C = dark blue; Sample 13,204 = sphere, 13,208 = diamond, 13,209 = diamond, 13,213 = square). For each treatment, there was a total of four samples. (A) Shows the microbial community differences at the feature level using Bray-Curtis; (B) shows differences at the feature level using Aitchison; (C) shows differences at the genus level using Bray-Curtis; and (D) shows differences at the genus level using Aitchison.

suggesting that distinct microbial profiles were obtained using either of the three methods. This observation corroborates with the findings of previous studies (*Weyrich et al., 2017*; *Ziesemer et al., 2015*). The median number of observed features in the 16S rRNA amplicon data were less than both the 16S enriched and UeWGA approaches, likely because the amplicons were too long to recover most of the taxa. It is interesting that the enrichment and the 16S amplicon methods recovered similar number of genera from the samples, since the enrichment method only recovers oral microbes. Nevertheless, the results make it clear that the UeWGA method identifies the greatest number of taxa and had less range of variation than either method, demonstrating that it should still be the preferred choice when surveying the microbial diversity of samples.

**Table 2 Compositional statistics for library method and hybridization temperature.**

| Taxonomic level | Beta-diversity metric | Variable | Statistical test | pseudo-F statistic | p-value | Sample size |
|---|---|---|---|---|---|---|
| **55° vs 65°** | | | | | | |
| Feature | Aitchison | Temperature | PERMANOVA | 0.00446 | 0.939 | 8 |
| Feature | Bray-Curtis | Temperature | PERMANOVA | 0.139371 | 0.798 | 8 |
| Genus | Aitchison | Temperature | PERMANOVA | 0.00967 | 0.803 | 8 |
| Genus | Bray-Curtis | Temperature | PERMANOVA | 0.047825 | 0.801 | 8 |
| **Enriched, Unenriched, and 16S** | | | | | | |
| Feature | Aitchison | LibraryType | PERMANOVA | 16.211176 | 0.001 | 42 |
| Feature | Bray-Curtis | LibraryType | PERMANOVA | 10.729735 | 0.001 | 42 |
| Genus | Aitchison | LibraryType | PERMANOVA | 1.898626 | 0.098 | 42 |
| Genus | Bray-Curtis | LibraryType | PERMANOVA | 6.39069 | 0.001 | 42 |
| **Enriched vs Unenriched** | | | | | | |
| Feature | Aitchison | LibraryType | PERMANOVA | 25.545544 | 0.001 | 28 |
| Feature | Bray-Curtis | LibraryType | PERMANOVA | 61.44512 | 0.001 | 28 |
| Genus | Aitchison | LibraryType | PERMANOVA | 10.654381 | 0.002 | 28 |
| Genus | Bray-Curtis | LibraryType | PERMANOVA | 5.766889 | 0.001 | 28 |

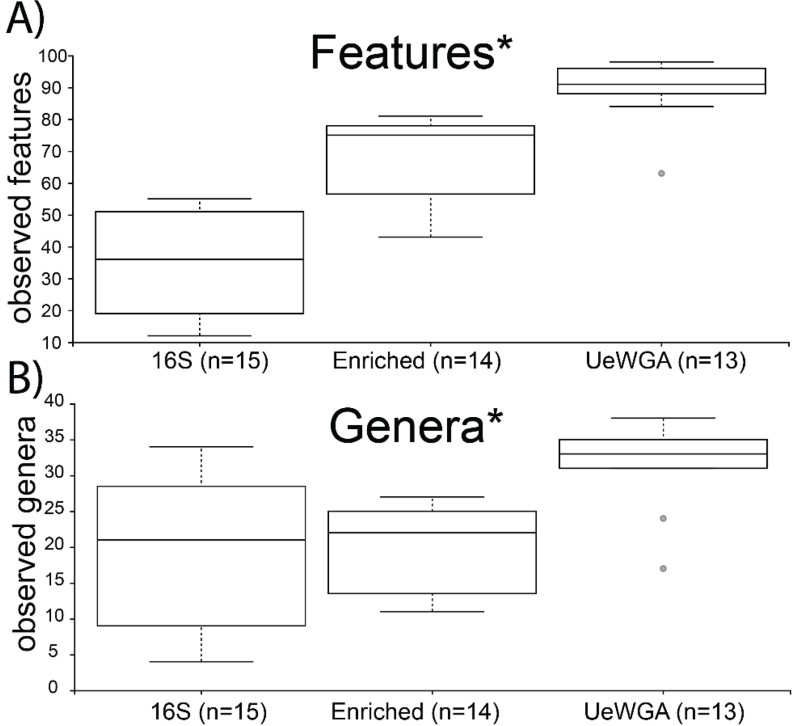

**Figure 4 Boxplot for group-specific observed features and genera distributions.** Kruskal-Wallis tests for all groups using features (*i.e.*, sequences identified at the lowest taxonomic rank) (A) and genera (B) were significant ($p < 0.05$) as indicated with an asterisk (*). The alpha diversity between each pairwise comparison was also significant (Table S6).

When comparing compositions across the methods, PCoA analysis of either Bray Curtis or Aitchison distances resulted in distinct clustering based on library type rather than the individual (PERMANOVA; $p$ = 0.001; Figs. 5A–5D), although the 16S enriched and UeWGA datasets clustered more closely together than the 16S amplicon data sets on Axis 1. This supports previous findings highlighting the strong contaminant biases present in ancient 16S rRNA gene amplified libraries (*Weyrich et al., 2017*). Biplots indicate that distinct clustering of the samples generated by 16S amplicon sequencing had higher abundances of contaminant taxa at the feature and genus level (*e.g.*, *Acientobacter, Comamonas, Comamonadaceae*, and *Eubacteriales*) (Figs. 5B and 5D), which has already been previously noted (*Ziesemer et al., 2015*).

We also explored differences between enriched 16S and UeWGA methods using a presence/absence approach. Taxa identified in the UeWGA data included *Bacteroides, Microbacter, Parabacteriodes, Desulfoplanes, Pseudomonas, Merismopedia, Blautia, Acholeplasma*, and *Candidatus Phytoplasma* (Fig. S5). These taxa are not commonly found in the oral cavity (*Chen et al., 2010*) and are likely present because of either laboratory or reagent contamination. It is possible that they were identified here because the minimum percentage needed for taxonomic assignment (minimum support percent 0.1%) for the reads of the UeWGA data is lower (*i.e.*, the minimum support percent for 3,000 assignments is (3,000*0.001 = 3)) than the 16S enriched data (minimum support percent for 30,000 (30,000*0.001 = 30)).

## Specific taxonomic differences between 16S rRNA enriched and UeWGA data sets

Given the known biases that 16S rRNA amplification methods have when applied to aDNA datasets, we removed samples generated from this method and proceeded to compare the compositional differences between the 16S enrichment and UeWGA methods (Table S7). A pairwise comparison based on Bray Curtis and Aitchison distances indicated that the microbial compositions were significantly different at the genus and feature level ($q$ = <0.05; Table 2). As expected, the UeWGA method identified more species (average of 64.8% of assigned reads at the species level *versus* 1.8% for the 16S enrichment method). Importantly, the UeWGA method could align 59% of reads, whereas only 0.8% could be aligned for the 16S enrichment method (Table S7). We next tested if there were differences in genus-level assignments between the two methods. The 16S rRNA gene enrichment method classified 20 genera that were not present in the UeWGA method, although these assignments each had a mean abundance <1% (Table 3). The UeWGA method had 18 genus identifications absent in the 16S rRNA gene enrichment method, and all but three of these each had a mean abundance of <1% (Table 3). Comparing these assignments to taxa found in the Human Oral Microbiome Database (HOMD), 9/20 were putatively oral for the 16S enrichment method, while 14/18 for the UeWGA method. Notably, the 16S enrichment method did not detect the common oral genus *Tannerella*, while the UeWGA not only identified this genus but also had a mean abundance of 3.42% across samples. Additionally, the putative oral genus *Olsenella* was not detected in the 16S enrichment method but had a mean abundance of 7.85% in the UeWGA data set. These findings

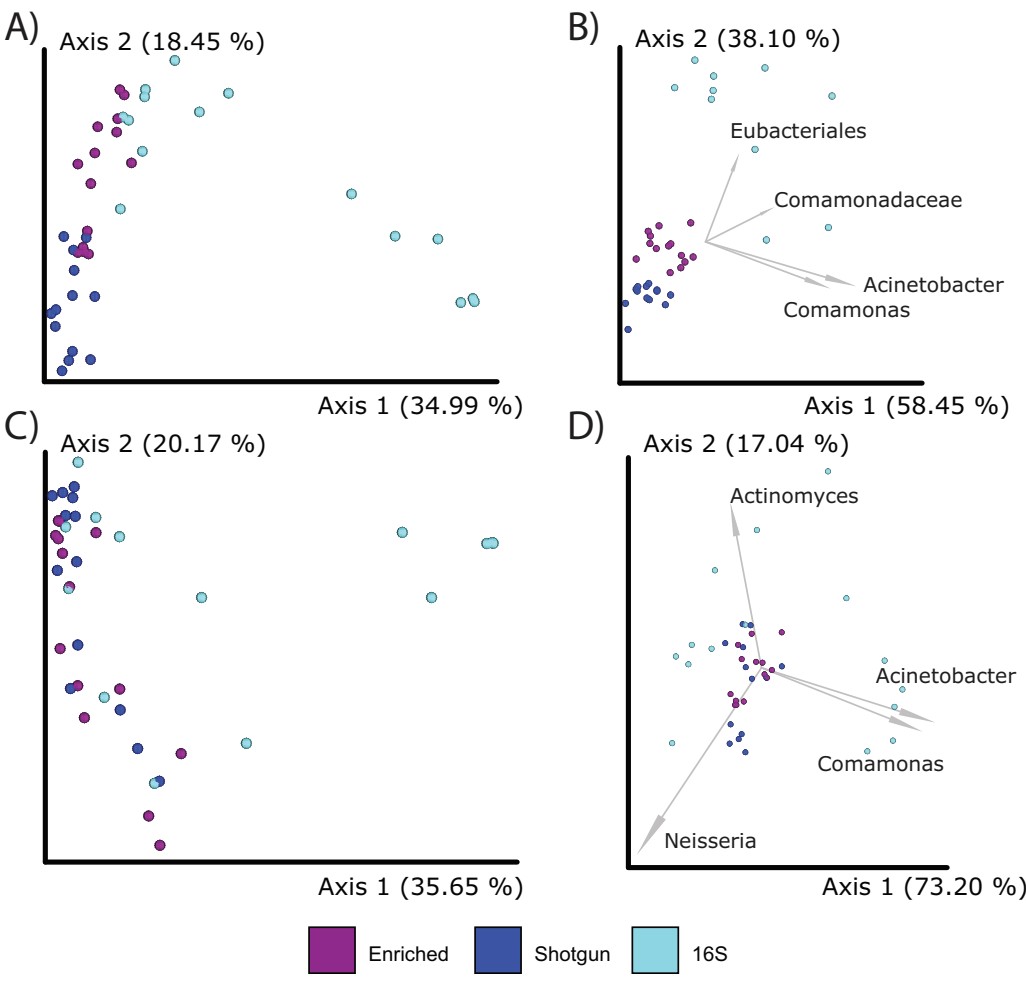

**Figure 5 PCoA plots comparing the microbial communities at different taxonomic levels and distance metrics.** Each sample is color coded to their respective LibraryType (Enriched = purple, UeWGA = dark blue, and 16S = light blue). For each treatment, there was a total of 15 samples. (A) The microbial community differences at the feature level using Bray-Curtis; (B) the differences at the feature level using Aitchisons; (C) the differences at the genus level using Bray-Curtis, and (D) shows differences at the genus level using Aitchisons.

suggest that rare or low abundant taxa are likely contributing to the minimal differences in composition between the two methods. As such, the 16S enrichment strategy missed key oral taxa that were identified in the UeWGA approach, meaning future should investigate how to improve probe designs for dental calculus samples to minimize the loss of important taxa.

## Comparison of cost estimates for enriched and shotgun methodologies

Given these observable compositional differences, we compared the costs associated with implementing 16S hybridization capture and UeWGA, given equal sequencing depth. When the sequencing data for each method was subsampled to the same depth, the 16S rRNA gene enrichment method assigned an average of 21.1% of sequences a taxonomic

**Table 3 Mean abundance of method-specific taxonomic assignments.**

| Taxon specific to 16S | Mean abundance | Taxon specific to shotgun | Mean abundance |
|---|---|---|---|
| **Peptococcus** | 0.90% | **Olsenella** | 7.85% |
| **Desulfobulbus** | 0.85% | **Tannerella** | 3.42% |
| **Mogibacterium** | 0.77% | **Parvimonas** | 1.19% |
| **Peptostreptococcus** | 0.41% | Dialister | 0.48% |
| **Fastidiosipila** | 0.39% | **Eikenella** | 0.45% |
| **Bergeylla** | 0.23% | **Atopobium** | 0.38% |
| Brachymonas | 0.22% | **Slackia** | 0.35% |
| Fusibacter | 0.19% | **Kingella** | 0.27% |
| Pelospora | 0.16% | Clostridium | 0.24% |
| Flavobacterium | 0.15% | **Peptoniphilus** | 0.17% |
| Roseburia | 0.14% | **Anaeroglobus** | 0.12% |
| Nocardioides | 0.13% | **Granulicatella** | 0.08% |
| **Paenibacillus** | 0.08% | **Solobacterium** | 0.06% |
| **Desulfovibrio** | 0.06% | **Oribacterium** | 0.05% |
| Polaromonas | 0.06% | **Eggerthia** | 0.05% |
| Candidatus Tammella | 0.05% | **Stomatobaculum** | 0.04% |
| Peptoclostridium | 0.04% | Chlorobium | 0.04% |
| Acetobacterium | 0.04% | Peptonaerobacter | 0.02% |
| **Pyramidobacter** | 0.02% | | |
| Verrucomicrobium | 0.02% | | |
| Sum of mean abundances | 4.92% | | 15.28% |

**Note:**
Bolded names represent taxa present in the Human Oral Microbiome Database.

classification to the genus and species level, while the UeWGA method assigned 34.5% of the sequences to a similar level. This represents a 13.4% increase in the number of reads assigned at the genus or species level for UeWGA over the 16S rRNA gene enrichment method, despite the increase in the percentage of reads assigned for the 16S rRNA gene enrichment method (56% for 16S rRNA gene enrichment, 41% for UeWGA). This suggests that while the 16S enrichment method can align and assign a higher percentage of sequences, these assignments are not as specific as the UeWGA method, and given the extra costs associated with the 16S rRNA gene enrichment method, the UeWGA method is currently the most cost-effective means of classifying genus or species level taxonomy in highly degraded ancient samples.

## DISCUSSION

Reconstructing ancient microbial communities is a challenging endeavor. Currently, only a few studies compare different methodologies and their impact on recovering ancient microbial communities (*Farrer et al., 2021*; *Velsko et al., 2018*; *Sarkissko et al., 2021*; *Eisenhofer & Weyrich, 2019*). Our study provides *in silico* simulations and empirical evidence to benchmark the effectiveness of applying an enrichment technique to recover 16S rRNA fragments from ancient shotgun metagenomes. While this new technique is not

as efficient at obtaining genus- or species-level resolution as UeWGA-based approaches, it still yielded a more accurate taxonomic profile (based on its similarity to the UeWGA data) than the 16S rRNA amplicon approach. It also recovered oral taxa that were not identified in the UeWGA data set. Therefore, the enrichment approach of 16S rRNA gene fragments could be useful to complement community reconstructions in addition to shotgun metagenomics approaches.

Microbiome research on modern populations report that longer lengths of the 16S rRNA can provide high-resolution taxonomic information (*Fuks et al., 2018*; *Martínez-Porchas, Villalpando-Canchola & Vargas-Albores, 2016*), and our simulated data indicated that indeed short 16S rRNA gene fragments did not contain sufficient information for species-level resolution. We were able to classify 98.4% of all simulated reads, with 53.2% of these being placed at the genus level. The resulting taxonomic classifications had no false-positive taxonomic assignments, and accurately recapitulated the true community. However, we did find that the 16S rRNA gene fragments for *E. coli* and *Y. pestis* 16S rRNA gene fragments were misclassified, which has been reported previously (*Chakravorty et al., 2007*; *Jovel et al., 2016*; *Warinner et al., 2017*). This misclassification may be due to insufficient resolution of the 16S rRNA gene to discriminate between these two species. Regardless, 16S rRNA gene fragments from these species were accurately classified at the family level, so not all taxonomic information was lost. By performing a reference sequence exclusion experiment, we found that our analytical method was robust to missing reference sequences in databases, which is currently a major issue for shotgun metagenomic methods. We also demonstrated that deamination insignificantly impacts the classification of short 16S rRNA gene fragments and that fragments as short as 30 bp could be assigned, supporting a previous assessment of alignment-based methods for aDNA (*Eisenhofer & Weyrich, 2019*). Overall, these simulations suggest that the alignment and classification of 16S rRNA fragments is robust to the characteristics of aDNA, allowing for its use in the reconstruction of ancient microbial communities in highly degraded samples.

To complement our simulated data and results, we developed a hybridization enrichment method to obtain 16S rRNA gene fragments from shotgun metagenomic libraries. For the 65 °C hybridization enrichment temperature tested, an average of 58% of sequenced DNA reads were 16S rRNA gene fragments, which represents a 334-fold increase over the same unenriched libraries. We also found that enriched libraries had longer read lengths than unenriched, which could be explained by shorter DNA fragments not having sufficient Watson-Crick hydrogen bonding sites to remain attached to probes at higher temperatures. Despite this difference in read lengths, we found that the taxonomic composition between the groups prepared at different temperatures (55° *vs* 65°) was minimally impacted ($p > 0.05$), suggesting temperature shifts result in minimal bias when enriching the 16S rRNA gene fragments of a shotgun metagenomic library.

Our hybridization enrichment technique also allowed us to empirically test the 16S rRNA gene diversity in metagenomic samples. While the enrichment method increases the sensitivity for detecting a few low abundant oral taxa in shotgun metagenomic libraries, the overall taxonomic composition remained similar, although there was a large difference in the number of putative 16S rRNA gene fragments (average of 3,231 for unenriched,

993,478 for enriched). This finding suggests that most 16S rRNA microbial diversity can be captured from unenriched shotgun metagenomic sequencing with as few as 3,000 16S rRNA gene fragments, corresponding to ~2,000,000 unenriched shotgun metagenomic sequences. This empirical verification adds further support to the microbial diversity captured in previous paleomicrobiology studies that used unenriched 16S rRNA gene fragments to classify ancient microbial communities (*Velsko et al., 2018*; *Weyrich et al., 2017*; *Ziesemer et al., 2015*).

This study empirically tested the specificity and sensitivity of taxonomic classifications derived from 16S rRNA gene fragment alignments and whole-genome alignments from aDNA libraries. We found that the UeWGA method has greater specificity than 16S rRNA gene fragment alignments, supporting *in silico* findings here and in a previous study (*Eisenhofer & Weyrich, 2019*), as well as in modern microbiome research (*Jovel et al., 2016*; *Ranjan et al., 2016*). While there were putatively oral taxa that were only identified in either the UeWGA and enrichment methods, UeWGA possessed a higher proportion of oral to contaminant taxa and was uniquely able to detect two relatively highly prevalent and abundant oral genera commonly found in dental calculus, *Tannerella* and *Olsenella*. Our *in silico* and empirical findings support the notion that the UeWGA approach is the most appropriate strategy for future aDNA work on dental calculus seeking to reconstruct microbial classifications. However, the enrichment of 16S rRNA gene fragments could be advantageous for studying poorly preserved samples, especially in contexts where environmental contamination is a major concern and obscures the endogenous signal. Applying the enrichment method on such samples could be useful for characterizing their taxonomic diversity and being able to perform phylogenetic analyses.

While we found similarities in the specificity of assignments and taxa identified between the UeWGA and 16S rRNA enrichment methods, the identification and abundance of a few taxa differed between the two approaches. Nevertheless, the overall taxonomic composition at the feature and genus level between the two methods were more similar compared to the 16S rRNA amplicon data. We propose that employing an enrichment method may only be appropriate when attempting to recover rare taxa, specific regions of interest, or when the aDNA preservation of the calculus samples is suboptimal. Applying an enrichment method in conjunction with UeWGA methods may offer a more comprehensive assessment when characterizing the microbial diversity within ancient dental calculus samples and be an alternative method when traditional UeWGA methods fail or are inaccessible. Moving forward, researchers may want to consider designing baits for specific oral taxa, such as *Anaerolineaceae* bacterium oral taxon 439, that could then improve phylogenomic reconstruction.

## CONCLUSIONS

In summary, our *in silico* simulations and empirical findings elucidate the benefits and limitations associated with 16S rRNA hybridization enrichment methods when compared to UeWGA-based methods for reconstructing ancient oral microbial communities. While the hybridization 16S enrichment method recovers less microbial taxa than the UeWGA approach, our results show how hybridization capture can still enhance surveying the

microbial diversity of ancient calculus samples. Specifically, it recovered genera that were not detected with the UeWGA approach. Another constraint for the 16S enrichment method is the *a priori* design of probes, which limits the ability to identify microbes that have yet been characterized. Future studies should consider developing new probe designs that include novel oral taxa that have been characterized in recent studies (*Klapper et al., 2023*; *Velsko et al., 2023*; *Handsley-Davis et al., 2022*). Adding these taxa into the probe design may improve characterizing the microbial diversity for poorly preserved samples where most of the DNA originates from contaminant sources. In this case, hybridization capture may be the better approach, as the UeWGA approach would recover more contaminant DNA than the 16S enrichment method. Nevertheless, we recommend that a UeWGA-based approach is the most optimal approach in characterizing the microbial diversity of ancient calculus samples in most cases. Because 16S rRNA gene fragments generally make up less than 0.05% of the DNA in shotgun metagenomic libraries (*Guo et al., 2016*), estimating the taxonomic diversity of ancient samples based solely on 16S rRNA data will likely give inaccurate diversity estimates. In addition to reconstructing microbial diversity, hybridization capture could be useful for phylogenetic reconstruction. Designing probes that target microbial regions of specific oral taxa of interest could improve coverage and phylogenomic analyses, as has been illustrated in ancient pathogen work (*Bos et al., 2011*; *Schuenemann et al., 2011*; *Spyrou et al., 2018*). In conclusion, our results offer valuable insights for the potential of incorporating hybridization capture strategies alongside shotgun metagenomic methods and offer promise for forthcoming investigations aiming to reconstruct ancient microbial communities.

## ACKNOWLEDGEMENTS

We thank Alison from Arbor Biosciences for help with the bait design used in the project.

### Funding

The work was also supported by an Australian Research Council Future Fellowship Award to Laura Weyrich (FT180100407). The funders had no role in study design, data collection and analysis, decision to publish, or preparation of the manuscript.

### Grant Disclosures

The following grant information was disclosed by the authors:
Australian Research Council: FT180100407.

### Competing Interests

The authors declare that they have no competing interests.

### Author Contributions

- Raphael Eisenhofer conceived and designed the experiments, performed the experiments, analyzed the data, prepared figures and/or tables, authored or reviewed drafts of the article, and approved the final draft.
- Sterling Wright performed the experiments, analyzed the data, prepared figures and/or tables, authored or reviewed drafts of the article, and approved the final draft.
- Laura Weyrich conceived and designed the experiments, authored or reviewed drafts of the article, and approved the final draft.

## Microarray Data Deposition

The following information was supplied regarding the deposition of microarray data:

Eisenhofer & Raphael (2019). Final 16S rRNA gene bait design. The University of Adelaide. Dataset. https://doi.org/10.25909/5cc11894b0cc2.

## Data Availability

The sequence reads are available at GenBank: PRJNA685265.

The code to replicate the analyses and figures are available at GitHub and Zenodo:

- https://GitHub.com/SterlingLWright/16S

- Sterling Wright. (2024). SterlingLWright/16S: Scripts for 16S Enrichment for Ancient Oral Microbiomes (microbiome). Zenodo. https://doi.org/10.5281/zenodo.10477035.

## Supplemental Information

Supplemental information for this article can be found online at http://dx.doi.org/10.7717/peerj.16770#supplemental-information.

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
