# Peer review of "Benchmarking a targeted 16S ribosomal RNA gene enrichment approach to reconstruct ancient microbial communities"

_PeerJ, doi:10.7717/peerj.16770_

## Round 0.1 · original submission · Major Revisions

Dear Dr. Eisenhofer and colleagues:

Thanks for submitting your manuscript to PeerJ. I have now received three independent reviews of your work, and as you will see, the reviewers raised some concerns about the research. Despite this, these reviewers are optimistic about your work and the potential impact it will have on research studying approaches to microbiome, metagenome and evolutionary analysis. Thus, I encourage you to revise your manuscript, accordingly, considering all the concerns raised by both reviewers.

Please work to better frame your research questions. There is a lack of clarity throughout the manuscript, as pointed out by the reviewers. There needs to be a more complete background provided in the Introduction, and a revisiting of this with your findings in the Discussion, emphasizing the impact of your contribution.

Importantly, please ensure that an English expert has edited your revised manuscript for content and clarity. Please also ensure that your figures and tables contain all the information that is necessary to support your findings and observations.

There are many comments by the reviewers that ask for more information on specific issues; please address these.

I look forward to seeing your revision, and thanks again for submitting your work to PeerJ.

Good luck with your revision,

-joe

**Language Note:** The Academic Editor has identified that the English language must be improved. PeerJ can provide language editing services - please contact us at copyediting@peerj.com for pricing (be sure to provide your manuscript number and title). Alternatively, you should make your own arrangements to improve the language quality and provide details in your response letter. – PeerJ Staff

·

Basic reporting

The paper presents a novel approach to the taxonomic characterization of ancient microbiomes, which is a rapidly growing field. The authors developed, tested, and applied a hybridization enrichment technique to selectively target 16S rRNA gene fragments from ancient dental calculus samples. The study is well-structured, and the methodology is clearly explained. However, in terms of reporting and clarity of manuscript there are few typos, grammatical errors and some additional background information that can be improved upon.

1) Line 36: "these assignments may not the same level of specificity as those achieved as unenriched methods" should be "these assignments may not achieve the same level of specificity as those achieved by unenriched methods."

2. Line 63: "This process leads to contaminant DNA from modern taxa being preferentially amplified which confounds microbial composition estimates." should be "This process leads to the preferential amplification of contaminant DNA from modern taxa, which confounds microbial composition estimates."

3. Line 143: "The resulting sequencing data was converted into the fastq format using Illumina's bcl2fastq software, before being demultiplexed and demultiplexed using AdapterRemoval2 based on unique P5/P7 barcode combinations (Schubert et al., 2016)." The word "demultiplexed" is repeated twice.

4. . The introduction could provide more background information on 1) the importance of studying ancient microbiomes and the challenges associated with this task, 2) Dental calculus and 3) Shotgun metagenomic data analysis for non-science folks.

Experimental design

The paper is well-structured and provides a comprehensive analysis of the proposed method. However, can you please address the following:

1) The methods section is quite dense and could be clarified further. It would be helpful to provide a more detailed explanation of the hybridization enrichment technique, including how it specifically targets 16S ribosomal RNA gene fragments.

Validity of the findings

The paper can be further improved if the following comments can be addressed:

1) While the paper does discuss some limitations, this could be expanded upon. It would be beneficial to discuss potential drawbacks of the method and how these might be addressed in future research.

2) The paper could provide a more detailed comparison of the proposed method with other existing methods. This would help to highlight the advantages and disadvantages of each approach.

3) The discussion should clearly state the implications of the research and suggest areas for future investigation. This helps to situate the research within the broader scientific discourse and indicates how it advances knowledge in the field.

4) The conclusion should succinctly summarize the main findings, their implications, and the significance of the research. It should be impactful and leave the reader with a clear understanding of the value of the study.

Reviewer 2 ·

Basic reporting

The authors have effectively employed a targeted 16S ribosomal RNA gene enrichment approach to reconstruct ancient microbial communities, demonstrating proficiency in their research. Nonetheless, the article exhibits several limitations, as discussed below.

1. The authors have made several misreferences throughout the article. Please review and correct these misreferences for clarity and accuracy in the manuscript.

a. In the legend of Figure 2, they incorrectly referred to the subplots as "Figure 4X."

b. In line 284, when discussing Bray-Curtis and Aitchison distances at the genus and features level, the authors mistakenly referred to Table 3 instead of Table 2.

c. In line 326, the reference to Figure S3 led to the incorrect supplementary figure, which could potentially confuse readers.

d. In line 343, there is ambiguity regarding whether the percentage is "% aligned" or "% not aligned" in the text. The description in the text does not match with the information in Table S6.

2. To enhance clarity in the manuscript, it's important to standardize the terminology used to describe the various sequencing approaches. In Figure 3, the authors presented three distinct library types: enriched, shotgun, and 16S. However, throughout the text, different terms like "16S enriched," "UeWGA," and "unenriched 16S" were used when referring to these library types. This inconsistency can potentially confuse readers as they attempt to correlate the discussion with the figures. It is advisable to employ consistent terminology to align the figure labels with the terms used in the accompanying text for improved comprehension.

3. To enhance the usability of supplementary materials for readers, it is advisable to consolidate all supplementary figures and tables into a single PDF file that include accompanying legends. Additionally, it is important to address any missing information in the supplementary figures. For instance, Figure S1 lacked a y-axis title, which is crucial for proper interpretation. These improvements will greatly benefit readers by providing a more comprehensive and accessible supplementary section.

Experimental design

Overall, the authors have provided a thorough account of their computational pipelines and statistical analyses. However, the introduction of several new terminologies and methods lacks comprehensive definition and discussion, potentially leaving the scientific community without sufficient guidance to replicate the results effectively.

a. In Figure 1 and Table 1, the authors conducted alignment statistics on simulated data, including a category called "LaBrana deamination." The text did not define LaBrana deamination entails and give the rationale behind its inclusion in the simulations.

b. In line 309, when referencing alpha diversity, the method of alpha diversity calculation remains unclear from the information provided in Table S5. It would be beneficial to include an equation or a more detailed explanation to elucidate the exact methodology used for calculating alpha diversity.

Validity of the findings

The authors have presented numerous comparisons and findings related to different sequencing approaches. However, in many instances, they have described phenomena without delving into a comprehensive explanation of the results and their implications, which could significantly diminish the paper's overall significance. The following areas could benefit from more extensive discussions:

a. In line 358, the authors noted that the specific 16S enrichment strategy resulted in the omission of key oral taxa, which is a substantial drawback for this method. It is crucial to elaborate on this omission and explore potential strategies to mitigate its negative impact. Are there approaches that could be employed to minimize the loss of important taxa when using the 16S enrichment method?

b. In line 403, the observation that deamination has little impact on the classification of short 16S rRNA gene fragments is intriguing and holds significant implications. A deeper exploration or the presentation of a hypothesis explaining why deamination has this limited effect on the classification of short 16S rRNA sequences would enhance the paper's scientific depth.

c. In Figure 4, it is evident that there is a considerable variance in the observed features and genera between the 16S and enriched methods compared to UeWGA. Additionally, the authors have highlighted that UeWGA outperformed both 16S and enrichment strategies in terms of taxa recovery at the feature and genus levels. To provide a more comprehensive understanding, it is essential to delve into the root causes of this observed discrepancy. What factors may contribute to these differences, and what are the broader implications for the choice of sequencing methods in similar studies? A detailed explanation would greatly enhance the clarity and significance of the findings.

Reviewer 3 ·

Basic reporting

Wright, et al. have tested the feasibility of 16S rRNA gene capture sequencing in ancient metagenome samples using simulations and real data captured with a new 16S gene-based capture array.

This paper could be interesting, but as it’s currently presented, it is difficult to follow the flow of the testing and comparisons that were done. This makes it difficult to see the value of a capture enrichment for 16S genes from ancient data, given that taxonomic identification based on 16S gene amplicon sequencing is inadvisable for ancient samples.

To start, an overview figure would help clarify the different parts, where they overlap and where they differ. I find it difficult to understand why different approaches were taken for data clean up and data analysis of different input data. Also, ordering the methods and results to match would help me keep track of the study as I go through.

I have questions regarding the methodological choices, and the justifications for them. For example,

1. Why does the introduction focus on sequence alignment for taxonomic identification but not mention faster, less memory-intensive approaches? Using only 16S rRNA genes is certainly faster than aligning all sequences, but it is well known that the resolution of 16S rRNA genes is low at the species level, and k-mer matching taxonomic assignment approaches are fast and have better species-level resolution. It’s not clear why it’s worth continuing to use the 16S rRNA gene for taxonomic classification in light of this information.

2. Why did the authors use a simulated community of only 19 species? And why include so many that are not oral taxa, although they are using ancient dental calculus as their reference?

3. Please explain why different datasets were processed differently, including rarefaction, database used, and taxonomic assignment program used.

4. Please specify the number of samples or libraries included in each set of analyses.

5. Why was the real damage pattern data simulated using a human sample, and not a bacterial species from an ancient calculus sample?

6. The results seem to indicate that all 16S rRNA gene-based approaches for calculus microbial community reconstruction are biased and should not be used, but this is not clear in the authors’ interpretation in the discussion. I’m uncertain whether there is any benefit to any of the 16S rRNA gene-based classification methods, whether pulled from shotgun data, amplicon sequenced, or captured. Could the authors clarify this in their final paragraph?

Experimental design

See comments in section 1.

Validity of the findings

See comments in section 1.

Additional comments

Some smaller additional points:
1. The reference to Yates, et al. 2021 on line 51 should be Fellows Yates, et al. 2021.
2. Supplemental Table 1 is mentioned several times in the text, but the information referenced there seems to be different at each mention. Please make sure the referenced information is consistent.
3. Please upload the capture data (and any other new data presented in this study) to SRA or ENA.
4. Please provide a list of accessions for the data that was used, both published and newly generated for this study.
5. Why have the authors referenced a metabolomics paper on line 430? Should this instead be the other Velsko, et al. paper in the reference list?
6. Table 1 presents data from which of the simulated data sets? Which damage level? How many repetitions were performed, what is the standard deviation?
7. The legend for Figure 2 mentions 4A, 4B, 4C, 4D. Please check this.
8. The 3rd dimension on the PCA plots doesn’t add additional information, but does make the plots more difficult to interpret. Can the authors change the plots to show only 2 dimensions?
9. What are the codes in the plots of Figure 2? Please indicate in a legend and in the text. Could the authors use both shape and color, so that the codes are indicated by one and temperature by the other, rather than adding circles and text to the plots?
10. The plot labels are inconsistent between Figures 3 and 4, but surely refer to the same groups? Can they be made consistent?
11. The significance stars on Figure 4 are difficult to interpret, and it’s unclear why the significant differences are referenced in Table S6 but not indicated in the plots. Please indicate the significant differences directly in the plots.
12. Please specify in Table 3 whether the 16S column refers to captured 16S data, or one of the other 16S approaches.

---

## Round 0.2 · accepted · Accept

Dear Dr. Eisenhofer and colleagues:

Thanks for revising your manuscript based on the concerns raised by the reviewers. I now believe that your manuscript is suitable for publication. Congratulations! I look forward to seeing this work in print, and I anticipate it being an important resource for groups studying approaches to microbiome, metagenome and evolutionary analysis.

Thanks again for choosing PeerJ to publish such important work.

Best,

-joe

·

Basic reporting

No comment

Experimental design

No comment

Validity of the findings

No comment

Additional comments

The authors have demonstrated a notable commitment to enhancing the quality of their manuscript by thoroughly addressing the suggestions and recommendations previously outlined. Their attention to detail and willingness to engage in a constructive dialogue through patient and comprehensive responses to my queries have significantly contributed to refining their work.

Therefore, it is with confidence and satisfaction that I offer my support for its publication, as the manuscript now presents a well-structured and valuable contribution to its respective field of study.